# Several Yeast Species Induce Iron Deficiency Responses in Cucumber Plants (*Cucumis sativus* L.)

**DOI:** 10.3390/microorganisms9122603

**Published:** 2021-12-16

**Authors:** Carlos Lucena, María T. Alcalá-Jiménez, Francisco J. Romera, José Ramos

**Affiliations:** 1Department of Agronomy (DAUCO-María de Maeztu Unit of Excellence), Edificio C-4, Campus de Rabanales CeiA3, Universidad de Córdoba, 14071 Córdoba, Spain; b42lulec@uco.es (C.L.); ag1roruf@uco.es (F.J.R.); 2Department of Agricultural Chemistry, Edaphology and Microbiology, Edificio C-6, Campus de Rabanales CeiA3, Universidad de Córdoba, 14071 Córdoba, Spain; maritrinialji97@gmail.com

**Keywords:** iron deficiency, *Hansenula polymorpha*, *Debaryomyces hansenii*, *Saccharomyces cerevisiae*, cucumber plants

## Abstract

Iron (Fe) deficiency is a first-order agronomic problem that causes a significant decrease in crop yield and quality. Paradoxically, Fe is very abundant in most soils, mainly in its oxidized form, but is poorly soluble and with low availability for plants. In order to alleviate this situation, plants develop different morphological and physiological Fe-deficiency responses, mainly in their roots, to facilitate Fe mobilization and acquisition. Even so, Fe fertilizers, mainly Fe chelates, are widely used in modern agriculture, causing environmental problems and increasing the costs of production, due to the high prices of these products. One of the most sustainable and promising alternatives to the use of agrochemicals is the better management of the rhizosphere and the beneficial microbial communities presented there. The main objective of this research has been to evaluate the ability of several yeast species, such as *Debaryomyces hansenii*, *Saccharomyces cerevisiae* and *Hansenula polymorpha*, to induce Fe-deficiency responses in cucumber plants. To date, there are no studies on the roles played by yeasts on the Fe nutrition of plants. Experiments were carried out with cucumber plants grown in a hydroponic growth system. The effects of the three yeast species on some of the most important Fe-deficiency responses developed by dicot (Strategy I) plants, such as enhanced ferric reductase activity and Fe^2+^ transport, acidification of the rhizosphere, and proliferation of subapical root hairs, were evaluated. The results obtained show the inductive character of the three yeast species, mainly of *Debaryomyces hansenii* and *Hansenula polymorpha*, on the Fe-deficiency responses evaluated in this study. This opens a promising line of study on the use of these microorganisms as Fe biofertilizers in a more sustainable and environmentally friendly agriculture.

## 1. Introduction

Iron (Fe) is an essential micronutrient for plants [1]. Iron deficiency is a major agronomic problem, present in approximately 30% of cultivated soils in the world [2,3]. Fe is very abundant in most soils, mainly in the Fe^3+^ form, although is poorly soluble and with low availability for plants, especially in calcareous soils with high pH [3,4]. This circumstance leads frequently to Fe deficiency and therefore to a significant decrease in the yield and quality of crops [5]. On the other hand, excessive Fe accumulation in plants can cause toxic effects [6,7]. For this reason, Fe acquisition should be tightly regulated.

Dicotyledonous (Strategy I) plants develop several Fe-deficiency responses, mainly in their roots, to cope with this constraint [8,9]. These responses, both morphological and physiological, are aimed to facilitate Fe mobilization and acquisition and fulfil the plant requirements of this element.

Within the morphological responses, Fe deficiency can cause an inhibition of primary root elongation, accompanied by the development of lateral roots [10,11,12]. Also, subapical root swelling, with large number of root hairs, can be developed to increase the contact surface of the root with the soil and improves Fe absorption [8,13]. The origin of these root hairs is epidermal and they are only developed in the subapical regions of the roots [8]. Within the physiological responses, dicot plants mobilize Fe from the soil through the release of protons and the subsequent acidification of the rhizosphere. This serves to solubilize Fe^3+^ and to enhance the reduction of Fe^3+^ to Fe^2+^ by means of a ferric reductase located in the plasma membrane of epidermal cells. The Fe^2+^ generated is then absorbed through a Fe transporter, also located in the plasma membrane of epidermal cells [8]. Some years ago, the genes encoding these proteins were identified in Arabidopsis, such as the ferric reductase *AtFRO2* [14], the Fe transporter *AtIRT1* [15], and the H^+^-ATPase *AtAHA7* [16]. Homologues of these genes have also been identified in other dicot plant species, such as tomato, pea, and cucumber [17,18,19,20]. The regulation of these Fe acquisition genes is mediated by several transcription factors (TFs). Two of the master regulators are the tomato FER protein, identified as a basic helix–loop–helix (bHLH) TF [21], and the Arabidopsis FIT protein (previously named bHLH29, FIT1, or FRU), a homologue of the tomato FER protein [6,22].

Once adequate Fe has been absorbed, Fe-deficiency responses need to be switched off to minimize Fe toxicity and energy costs. Their regulation is not fully understood, but several hormones and signaling molecules, such as auxin, ethylene (ET), and nitric oxide (NO), which increase their production in Fe-deficient roots, have been proposed to participate in the activation of most responses in dicot plants [8,23,24]. Auxin, ET, and NO are closely interrelated in a complex manner since each one can affect the production and/or distribution of the other ones [23,25].

Since the middle of the 20th century, there has been a great increase in world agricultural production, sometimes associated with the abusive use of chemical fertilizers. This has caused serious environmental problems, such as eutrophication of inland waters, loss of biodiversity, and, in addition, the degradation of the cultivated soil itself [26,27]. For this reason, it is necessary to find new strategies to feed plants with less dependence on chemical products [28].

In this sense, one promising approach is a better management of the rhizosphere and its associated microbial communities in soils [29,30]. There are rhizosphere-beneficial microbes that can improve the assimilation of nutrients by the plants, which can be used as a successful and sustainable strategy for fertilization in modern agriculture, avoiding exclusive dependence on chemical fertilizers.

As previously mentioned, to deal with Fe deficiency, plants develop morphological and physiological responses, mainly in their roots, aimed at facilitating iron acquisition [6,8,31]. Despite these responses, in many cases it is also necessary to apply Fe fertilizers to correct this deficiency in the field. The most common practice is the application of Fe chelates to the soil, which are generally expensive and consequently restricted to crops with high added value [3]. Another way to fight against Fe deficiency is the use of plant genotypes that are more efficient in Fe assimilation. However, different results obtained using sterile soils have shown that, even with these genotypes, the cooperation of the microorganisms of the rhizosphere is necessary for an adequate Fe acquisition [30,32]. Several studies, carried out with beneficial fungi or bacteria, have shown that its application to soils can improve Fe nutrition in plants [33,34,35,36,37,38,39,40,41]. However, nothing is known to date about the possible role that yeasts may play in the Fe nutrition of plants. Recently, [42] have characterized, for the first time, the role of certain strains of the yeast *Debaryomyces hansenii* in the detoxification of arsenic in rice plants. Furthermore, in that study the authors also observed that, apart from mitigating the stress caused by arsenic, inoculation with the yeast improved both the nutritional status and the growth of rice plants. In recent years it has been found that some rhizosphere microorganisms (fungi and bacteria) can induce physiological and morphological responses in the roots of dicot plants similar to those induced by Fe deficiency [29,32,35,43,44,45,46,47,48,49,50]. Some of these rhizosphere microorganisms are also capable of triggering an induced systemic response (ISR) to combat pathogen and insect attacks. This observation suggests that both processes (ISR and Fe-deficiency responses) could be closely interconnected, probably because they share similar signaling pathways, and opens the way to the use of beneficial rhizosphere microorganisms to improve plant Fe nutrition [29,44,45,50,51].

It is therefore important to expand and optimize the list of possible beneficial microorganisms that can help plants to assimilate Fe and, in this way, to minimize the use of chemical fertilizers. Here, we describe and evaluate yeast species that have the capacity to induce Fe-deficiency morphological and physiological responses and can help plants to acquire Fe. The yeast species analyzed have been *Debaryomyces hansenii*, the nitrate *Hansenula polymorpha*, and *Saccharomyces cerevisiae*, which have been applied to cucumber plants. *D. hansenii* was selected because is an abundant yeast in soil and salty environments. In addition, it is a yeast very tolerant to several abiotic stress factors. *H. polymorpha* was chosen because, in addition of being abundant in soil, it is a model to study mineral nutrition in plants since, for example, it is a nitrate-assimilating yeast. Finally, *S. cerevisiae* was used just because it is a research model and most of the information in yeast has been studied in *Saccharomyces*.

## 2. Materials and Methods

### 2.1. Plant and Microbial Materials

Cucumber plants (*Cucumis sativus* L. cv Ashley) were used.

As inoculum, three laboratory nonpathogenic strains of the yeast species *Debaryomyces hansenii* (CBS767, wild-type, Netherland collection; Ramos-Moreno et al. [52]), *Hansenula polymorpha* (NCYC495 leu2 ura3 strain; Cabrera et al. [53]), and *Saccharomyces cerevisiae* (DB746 MATalpha his3delta1 leu23 leu2112 ura352 trp1289; Serra-Cardona et al. [54]), were used.

### 2.2. Growth Conditions

Several experiments were carried out with cucumber plants grown in hydroponic culture in a growth chamber under controlled conditions, with a photoperiod of 14 h, temperature of 22–24 °C during the day and 18–20 °C during the night, relative humidity of 60–70%, and radiation of 200 µmol m^−2^ s^−1^ provided by fluorescent tubes “Sylvania Cool White VHO”.

The nutrient solution R&M [11], modified according to the different treatments, had the following composition: Ca(NO_3_)_2_ 2 mM, K_2_SO_4_ 0.75 mM, KH_2_PO_4_ 0.5 Mm, MgSO_4_ 0.65 mM, KCl 50 μM, H_3_BO_3_ 10 μM, MnSO_4_ 1 μM, CuSO_4_ 0.5 μM, ZnSO_4_ 0.5 μM, (NH_4_)_6_Mo_7_O_24_ 0.05 μM, and Fe-EDDHA 20 μM.

Different treatments were applied to study the effects of the three yeast strains on some Fe-deficiency responses in cucumber plants. Each experimental design was planned to monitor the effects of yeasts on plants over the four days after the yeasts were applied.

Seeds of cucumber (*Cucumis sativus* L. cv. Ashley) were germinated in the dark within papers moistened with 5 mM CaCl_2_. After 2–3 d, the seedlings were transferred to a plastic mesh held over a half-strength nutrient solution, and kept in the dark for 2 d. Cucumber seedlings were then transplanted individually to 70 mL plastic vessels containing continuously aerated nutrient solution with 20 µM FeEDDHA. After 2–3 d in this nutrient solution, cucumber plants were transferred to the different treatments.

### 2.3. Inoculum Preparation

The yeasts were grown in complex liquid YPD medium composed of 1% yeast extract, 2% peptone, and 2% glucose. Cultures were inoculated in flasks at absorbance at 600 nm (A_600_) 0.05 and incubated at 26 °C for *D. hansenii*, 37 °C for *H. polymorpha*, or 28 °C for *S. cerevisiae*. One to two days later, the yeasts reached the middle exponential growth phase and then were collected by centrifugation and washed twice with sterile cold water. Finally, cells were counted in a Neubauer chamber and yeast cells were resuspended in the plant nutrient solution to reach the desired cell concentration.

### 2.4. Treatments

Experiments were carried out with 14-d-old cucumber plants, which had been grown with a complete nutrient solution in hydroponic medium. These plants were subjected to Fe deficiency for 4 days and some of them were inoculated with yeasts. Four treatments were considered for each plant species: -Fe plants, -Fe plants inoculated with *D. hansenii*, -Fe plants inoculated with *H. polymorphous*, and -Fe plants inoculated with *S. cerevisiae* (the concentration of each yeast in the medium was 10^7^/mL medium). The inoculation was carried out by adding the corresponding volume of inoculum, depending on the concentration obtained for each yeast, to the nutrient solution in which each treated plant was found. A scheduled collection of plants (6 control and 6 inoculated) was carried out at 24, 48, 72, and 96 h after treatments. On each day of collection, the ferric reductase activity, root hairs proliferation, and the acidification of the nutrient solution (pH) were determined.

In addition, another experiment was carried out under same circumstances described above but under Fe-sufficient conditions (40 µM Fe). The acidification location using agar plates and root hairs proliferation were measured in this experiment.

### 2.5. Determinations

Two enzymatic activities activated by dicot plants under Fe deficiency were evaluated: ferric reductase activity and proton pump activity.

Ferric reductase activity was evaluated by determining the capacity of roots to reduce Fe^3+^ to Fe^2+^, which is then chelated by Ferrozine.

The proton pump activity, due to H^+^ -ATPases, was evaluated by determining the pH of the nutrient solution and also by using acidification localization techniques on bromocresol purple agar plates.

In addition, a morphological response induced by dicot species under Fe deficiency was analyzed: the proliferation of root hairs in the subapical regions of the roots. For this, roots were stained with toluidine blue and observed with a stereoscopic microscope.

• Ferric reductase activity

Intact plants were pretreated for 30 min in plastic vessels with 50–70 mL of a nutrient solution without micronutrients, pH 5.5, and then placed into 50–70 mL of a Fe^3+^ reduction assay solution for 30 min. This assay solution consisted of nutrient solution without micronutrients, 100 µM Fe^3+^-EDTA and 300 µM Ferrozine, pH 5.0 (adjusted with 0.1 N KOH). The environmental conditions during the measurement of Fe^3+^ reduction were the same as the growth conditions described above. The ferric reductase activity was determined spectrophotometrically by measuring the absorbance (562 nm) of the Fe^2+^–Ferrozine complex and using an extinction coefficient of 29,800 M^−1^ cm^−1^ [55]. After the reduction assay, roots were excised and weighed, and the results were expressed on a root fresh-weight basis. In some treatments, the location of the ferric reductase activity along the roots was visualized in agar plates with ferric reduction assay solution [55].

• Nutrient solution acidification

A daily monitoring of the pH of the nutrient solution, in which the plants of the different treatments were grown, was carried out by means of a portable pH meter model HI 8424.

In addition, agar plates with the pH indicator bromocresol purple were prepared, with the aim of locating the regions of roots where acidification occurred. This determination was used to find possible local inductions masked by registering the pH of the entire solution.

• Subapical root hairs

One of the main morphological Fe-deficiency responses is the formation of root hairs in the subapical regions of the roots, where most of the Fe-deficiency physiological responses are found. Daily, visual monitoring was performed at the time of the treatments. When differences between treatments were appreciated, roots were sampled for subsequent staining and digitization. Toluidine blue was used for root staining. After 10 min, the excess of dye was removed with plenty of water for later observation under a stereomicroscope equipped with a built-in camera.

• qRT-PCR Analysis

Roots were ground to a fine powder with a mortar and pestle in liquid nitrogen. Total RNA was extracted using the Tri Reagent solution (Molecular Research Center, Inc., Cincinnati, OH) according to the manufacturer’s instructions. M-MLV reverse transcriptase (Promega, Madison, WI) was used to generate cDNA with 3 μg of total RNA from roots as template and random hexamers or oligo dT(20) as primers. Prior to cDNA synthesis, RNA was treated with DNAse to eliminate possible contamination by genomic DNA, and was DNase-inactivated later by adding 50 mM EDTA. Negative controls included all reaction components except the M-MLV enzyme. One tenth of each RT reaction was used as a PCR template.

The study of gene expression by qRT-PCR was performed using a qRT-PCR Bio-Rad (CFX connect), and the SYBR Green Bio-RAD PCR Master Mix, following the manufacturer’s instructions. *SAND1* and *YLS8* genes were used as reference genes to normalize the results of the qRT-PCR. The Pfaffl method was used to calculate the relative expression levels. Primer pairs for cucumber genes were designed as follows: (5′-3′) CsFRO1F (ATA CGG CCC TGT TTC CAC TT); CsFRO1R (GGG TTT TGT TGT GGT GGG AA); CsIRT1F (GCA GGT ATC ATT CTC GCC AC); CsIRT1R (ATC ATA GCA ACG AAG CCC GA); CsHA1F (GGG ATG GGC TGG TGT AGT TT); CsHA1R (TTC TTG GTC GTA AAG GCG GT).

### 2.6. Yeast Viability

A daily monitoring of the viability of the yeasts used in this work was carried out during the four days of the treatments. For this, several plants were chosen from each treatment and samples were taken every day from the nutrient solutions where plants grew. These samples were seeded in plates with the different specific culture media used for the growth of each yeast, mentioned above. It was found that the yeasts remained alive and active throughout the experiments.

### 2.7. Statistical Analysis

All experiments were repeated at least twice and representative results are presented. Both values of ferric reductase activity and pH of the nutrient solution represent the mean ± SE of six replicates. The values of qRT-PCR represent the mean ± SE of three independent biological replicates and two technical replicates. Within each day, ** or *** indicate significant differences (*p* < 0.01 or *p* < 0.001) in relation to the control treatment, using one-way analysis of variance (ANOVA) followed by a Dunnett’s test. Data were analyzed using STATISTIX 10 (SPSS (version 25), IBM, Armonk, NY, USA).

## 3. Results

### 3.1. Effect of Debaryomyces hansenii (Dh), Hansenula polymorpha (Hp), and Saccharomyces cerevisiae (Sc) on Ferric Reductase Activity and FRO1 Expression

Inoculation of -Fe treated cucumber plants with the yeast *Dh* caused a greater induction of ferric reductase activity than that achieved by noninoculated plants (Figure 1a). The highest induction was reached on the fourth day, just when the control plants lost their induction capacity. Something similar happened when plants were inoculated with *Hp* or *Sc* (Figure 1c,e). Cucumber plants inoculated with any of the yeast species mentioned maintained the same level of induction of ferric reductase activity during the second, third, and fourth day after treatments.

According to the last results, the expression of *CsFRO1*, the gene coding for the ferric reductase, was highly induced by the presence of any of the three yeast species just at the end of the experiments, regarding what happened with control plants (Figure 1b,d,f). These higher inductions were found at the fourth day after inoculation, just at the moment in which control plants grown without Fe lost the induction capacity of the *CsFRO1* expression.

### 3.2. Effect of Debaryomyces hansenii (Dh), Hansenula polymorpha (Hp) and Saccharomyces cerevisiae (Sc) on pH of the Nutrient Solution and HA1 Expression

Two methods were used for pH determination in this study. A daily monitoring of the nutrient solution pH used for the cultivation of the plants during the treatment period was carried out. Results are presented in Figure 2. Another additional methodology was used for confirming results obtained by the first method. For this, plates with agar and bromocresol purple were prepared to determine the location of acidification in cucumber roots grown in presence of each yeast under Fe-deficiency conditions (data not shown).

The inoculation with *Dh* or *Hp* induced higher acidification of the medium after three days of treatment (Figure 2). In both cases, higher significant differences (*p* < 0.001) between inoculated and noninoculated plants, were achieved after four days of treatments (Figure 2a,c). However, inoculation with Sc did not cause any inducing effect of acidification of the medium in cucumber plants (Figure 2e).

In relation to the expression of the gene that codes for the plasma membrane H^+^-ATPase in cucumber plants (*CsHA1*), our results demonstrated that the presence of *Dh* or *Hp* induced its expression considerably from the second day after inoculation respective to control plants (Figure 2b,d). No effect was found on the gene expression by the inoculation with *Sc*, under our experimental conditions (Figure 2f).

### 3.3. Effect of Debaryomyces hansenii (Dh), Hansenula polymorpha (Hp) and Saccharomyces cerevisiae (Sc) on Iron Transporter Gene IRT1 Expression

The early and forceful effect of inoculation with *Dh* or *Hp* on *CsIRT1* (iron transporter gene) expression, one of the most important genes involved in Fe acquisition, is shown in Figure 3a,b. Inoculation with *Dh* or *Hp* enhanced *CsIRT1* expression after two days of treatment, respective to control plants, showing high significative differences. No effect was found on *CsIRT1* expression by the inoculation with *Sc*, under our experimental conditions (Figure 3c), which is in agreement with results in Figure 2f.

### 3.4. Effect of Debaryomyces hansenii (Dh), Hansenula polymorpha (Hp), and Saccharomyces cerevisiae (Sc) on the Development of Subapical Root Hairs

The results obtained show that both *Dh* and *Hp* induced the development of subapical root hairs from the first day of treatment (Figure 4). However, there was not inducing effect by *Sc* (Figure 4). The development of subapical root hairs begun to be subtly seen from the fourth day of treatment in -Fe plants. Inoculation with *Dh* or *Hp* considerably increased this response in –Fe plants. The proliferation of root hairs was observed, practically, from the first day and was more evident on the fourth day after inoculation (Figure 4).

In order to get additional information about this important effect, the three yeast species were inoculated in cucumber plants grown under Fe-sufficient conditions. Under these conditions (Figure 5), we could verify that the inoculation with any of the three yeast species had a similar effect to the one observed under Fe-deficiency conditions. *Dh* and *Hp* induced their proliferation, though perhaps not as soon as they did on –Fe plants. *Sc* did not have any inducing effect on the proliferation of hairs in plants cultivated with Fe, as also happened on plants cultivated without Fe (Figure 4).

## 4. Discussion

In recent years it has been found that some beneficial rhizospheric microbes (rhizobacteria and rhizofungi) can induce physiological and morphological responses in the roots of dicot plants similar to the ones induced under Fe deficiency, thus facilitating Fe acquisition [29,35,40,41,50,51,56,57]. However, the role of yeasts on the induction of Fe-deficiency responses (to our knowledge) has not been described yet. Therefore, it is important to analyze whether yeasts are also involved in the induction of these responses under Fe deficiency.

Dicotyledonous plants respond to Fe deficiency by increasing the ability to reduce Fe^3+^, the most common form of Fe in most soils, to Fe^2+^, the form in which they acquire it [4,9,58]. This reduction is mediated by a ferric reductase enzyme whose optimum pH for its activity is in the range of 4 to 5 [59,60]. Acidification of the rhizosphere is another of the main responses that dicot plants activate under Fe-deficiency conditions [18], as well as the enhancement of the Fe^2+^ transport and the development of subapical root hairs [8]. The objective pursued by the plants with the implementation of these responses is to increase the solubility and bioavailability of Fe in the medium [8,9,31,50]. These Fe-deficiency responses are switched off once plants acquire enough Fe and are tightly coordinated and regulated by hormones and signaling molecules, such as ethylene and nitric oxide [14,61,62].

As shown in Figure 1a,c,e, the inoculation with the yeasts *Dh*, *Hp*, and *Sc* caused a greater induction of the ferric reductase activity than that achieved in -Fe plants just at the end of the experiments, and clearly correlated with the induction of *CsFRO1* gene expression (Figure 1b,d,f). As observed, when the induction levels drop in the control plants, the inoculated plants maintain the same high level reached during the previous two days. This prolonged induction of the ferric reductase and *CsFRO1* expression was observed with the three yeast species used in this research (Figure 1) and is a very interesting effect to take into account. Other authors have obtained a similar increase in the reductase activity in pear plants (*Pyrus communis*) inoculated with different plant growth-promoting bacteria (*Alcaligenes* spp., *Agrobacterium* spp., *Staphylococcus* spp., *Bacillus* spp., and *Pantoea* spp. [38]), or in tomato plants inoculated with *Trichoderma* sp. fungi [49]. However, until now, nothing was known on the effect that yeasts could have on ferric reductase activity. This opens a very interesting new line of study in order to discover the role that yeasts could play as inducers of Fe-deficiency responses and, consequently, as Fe biofertilizers.

The enhancement of the acidification response obtained with both *Dh* and *Hp* (Figure 2a,c) has also been described for other microorganisms. Thus, Zhang et al. [35], Pii et al. [30], and Ipek et al. [38] observed a greater Fe availability in soils inoculated with various growth-promoting bacteria (*Alcaligenes* spp., *Agrobacterium* spp., *Staphylococcus* spp., *Bacillus* spp. and *Pantoea* spp.) and pointed out that such an increase was due to a lower pH of the substrate derived from organic acids released by rhizobacteria. Our results obtained with *Dh* and *Hp* show an enhanced level of acidification on the third and fourth day of its presence in the medium (Figure 2a,c). That higher acidification was related to the enhancement of *CsHA1* expression, even one day earlier (since the second day after treatments), as shown in Figure 2b,d.

It should be noted that two of the three yeast species studied in this work (*Dh* and *Hp*) exert an acidification-inducing effect also on plants grown under Fe-sufficient conditions, where plants would not need to activate this response (data not shown). These results show for the first time that these yeasts can play a very important role in the induction of a response that favors Fe solubilization and acquisition [8].

Another gene involved in the Fe-acquisition strategy, the iron transporter gene *CsIRT1*, increased its expression in the presence of *Dh* or *Hp* (Figure 3a,b). These results confirm that the effects of both yeast species are generalized on the expression of genes encoding different Fe-deficiency responses.

It is in the root that the greatest changes caused by Fe deficiency occur, such as the inhibition of its elongation and the development of lateral roots [12]. In the subapical zones of the young roots under Fe deficiency, thickening of the root with great proliferation of root hairs originate [11]. With the formation of these root hairs, the contact surface of the root with the medium increases and, therefore, the probability of finding the nutrients increases [63,64]. Various authors have shown that microorganisms can play an essential role both in the formation and proliferation of root hairs [65].

The role that yeasts play on the formation of root hairs in the subapical zone of the roots has not been described to date by any author. Therefore, in this work we analyzed whether yeasts are also involved in the induction of morphological responses to Fe deficiency, as occurs with the physiological ones. The analysis was carried out in cucumber plants grown with or without Fe.

Cucumber, among other plant species, such as tomato, develop subapical root hairs under Fe deficiency [9].

Our results show that both *Dh* and *Hp* induce the development of subapical root hairs from the first day of treatment (Figure 4) even under Fe-sufficiency conditions (Figure 5). However, there is not any inducing effect by *Sc*. These modifications in the root architecture have also been observed with other microorganisms. Thus, Delaporte [66] reported a greater proliferation of root hairs in strawberry plants (*Fragaria ananassa*) inoculated with *Azospirillum brasilense* and *Gluconacetobacter diazotrophicus*. However, never before the effect of yeasts on the development of subapical root hairs been described. Recently, Kaur et al. [42] studied the effects of *D. hansenii* on plants but focused on the detoxification of arsenic in rice plants.

*Dh* and *Hp* can induce the formation of root hairs even in conditions of Fe sufficiency. This suggests that these microorganisms, with the capacity to induce several Fe-deficiency responses, could be used as Fe biofertilizers and, in this way, to reduce the use of chemically synthesized products.

In conclusion, our results indicate that at least two of the three yeast species studied (*Dh* and *Hp*) are potential candidates to be considered within the group of beneficial microorganisms capable of helping dicotyledonous plants for Fe nutrition. Both yeast species greatly induce several key Fe-deficiency responses and could be used in environmentally friendly agronomic fertilization techniques in substitution of the synthetic chelates used to fight against Fe chlorosis.

## Figures and Tables

**Figure 1 microorganisms-09-02603-f001:**
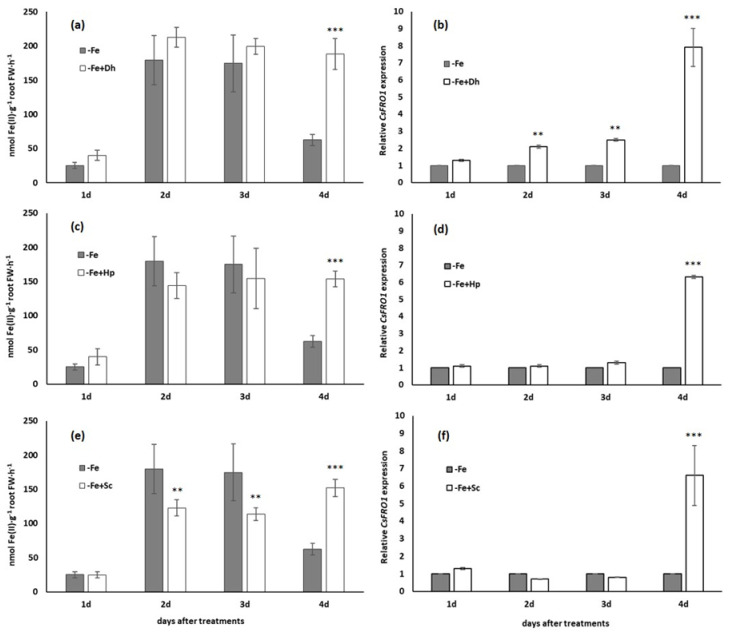
Time course of ferric reductase activity and *CsFRO1* expression in cucumber plants along the 4 days of treatments. Treatments: nutrient solution without Fe (-Fe) and inoculation with the different yeast species, *Debaryomyces hansenii* (−Fe+Dh) (**a**,**b**), *Hansenula polymorpha* (-Fe+Hp) (**c**,**d**), or *Saccharomyces cerevisiae* (−Fe+Sc) (**e**,**f**). The inoculation was carried out the same day Fe-deficiency treatment was applied. The data of ferric reductase activity are given as means ± SE (*n* = 6) while the data of *CsFRO1* expression represent the mean ± SE of three independent biological replicates and two technical replicates. Within each day, bars with ** or *** indicate significant differences (*p* < 0.01 or *p* < 0.001) in relation to the -Fe treatment according to the Dunnett’s test.

**Figure 2 microorganisms-09-02603-f002:**
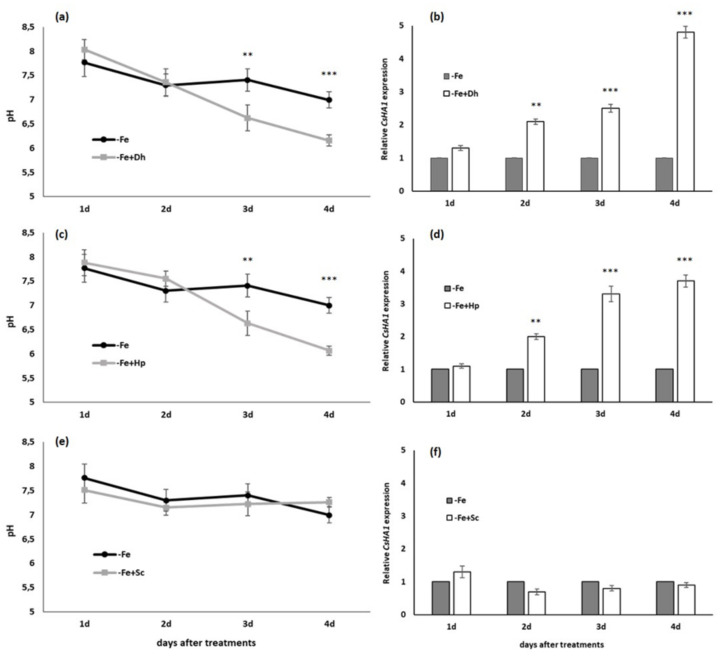
Time course of the pH of the nutrient solution and *CsHA1* expression in cucumber plants over the 4 days of treatments. Treatments: (control) without Fe (-Fe) and (inoculated) inoculation with different yeasts, *Debaryomyces hansenii* (−Fe+Dh) (**a,b**), *Hansenula polymorpha* (−Fe+Hp) (**c,d**), or *Saccharomyces cerevisiae* (−Fe+Sc) (**e,f**). The inoculation was carried out on the same day that Fe-deficiency treatment was applied. Data of pH are given as means ± SE (*n* = 6). The data of *CsHA1* expression represent the mean ± SE of three independent biological replicates and two technical replicates. Within each day, bars with ** or *** indicate significant differences (*p* < 0.01 or *p* < 0.001) in relation to the –Fe treatment according to the Dunnett’s test.

**Figure 3 microorganisms-09-02603-f003:**
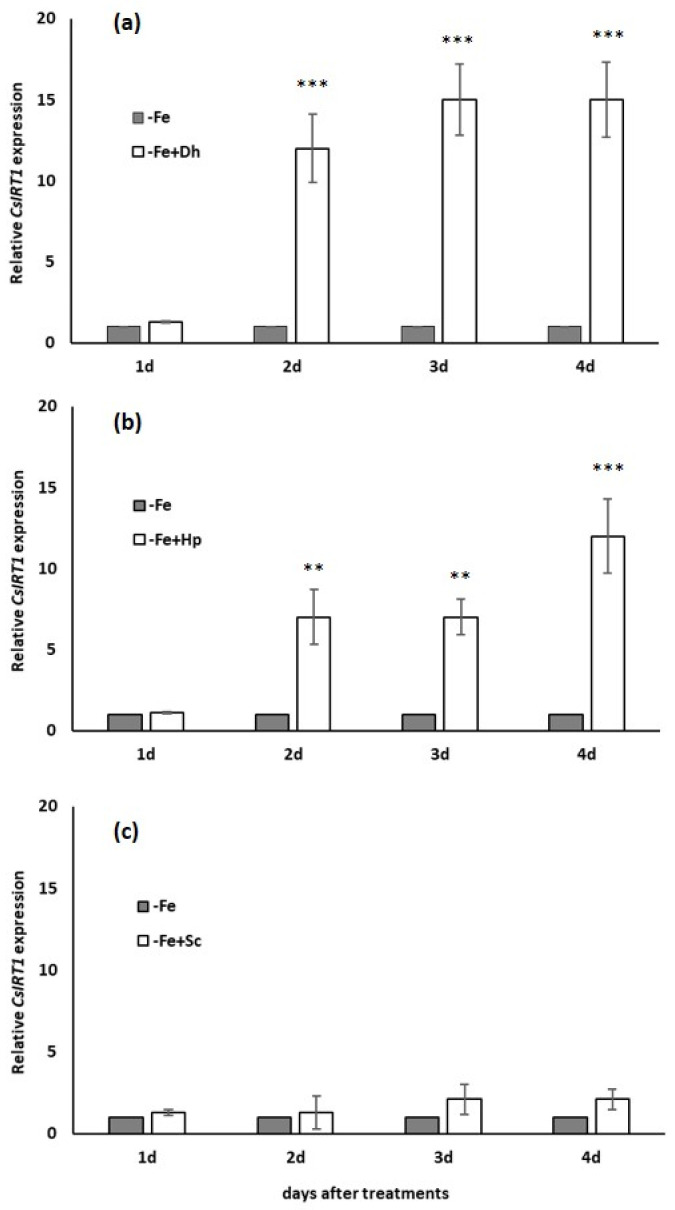
Time course of *CsIRT1* expression in cucumber plants over the 4 days of treatments. Treatments: (control) without Fe (−Fe) and (inoculated) inoculation with different yeasts, *Debaryomyces hansenii* (−Fe+Dh) (**a**), *Hansenula polymorpha* (−Fe+Hp) (**b**), or *Saccharomyces cerevisiae* (−Fe+Sc) (**c**). The inoculation was carried out the same day Fe-deficiency treatment was applied. Data *CsIRT1* expression represent the mean ± SE of three independent biological replicates and two technical replicates. Within each day, bars with ** or *** indicate significant differences (*p* < 0.01 or *p* < 0.001) in relation to the –Fe treatment according to the Dunnett’s test.

**Figure 4 microorganisms-09-02603-f004:**
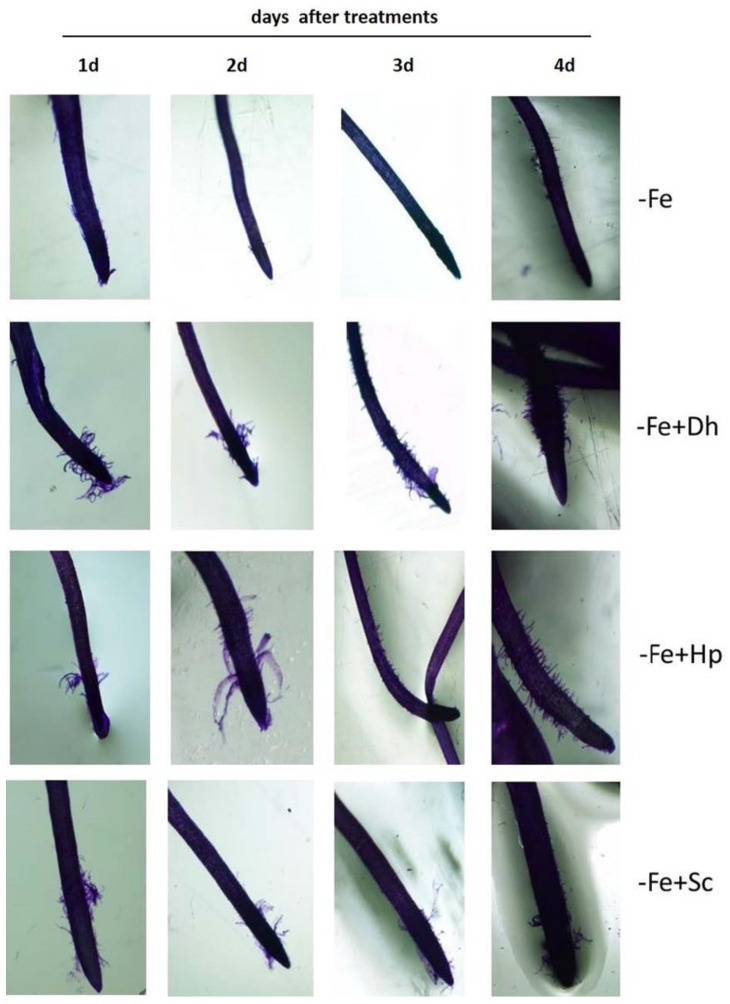
Analysis of development of subapical root hairs in young roots of cucumber plants grown without Fe (–Fe) or without Fe and inoculated with any of the three yeasts studied in this work: *Debaryomyces hansenii* (−Fe+Dh), *Hansenula polymorpha* (−Fe+Hp), or *Saccharomyces cerevisiae* (−Fe+Sc). The inoculation with the yeasts was carried out the same day the Fe-deficiency treatment was applied. Roots were observed 1, 2, 3, and 4 days after treatments (1d, 2d, 3d, and 4d). The images were obtained using a stereoscopic microscope with camera system after having stained the roots with toluidine blue for 10 min.

**Figure 5 microorganisms-09-02603-f005:**
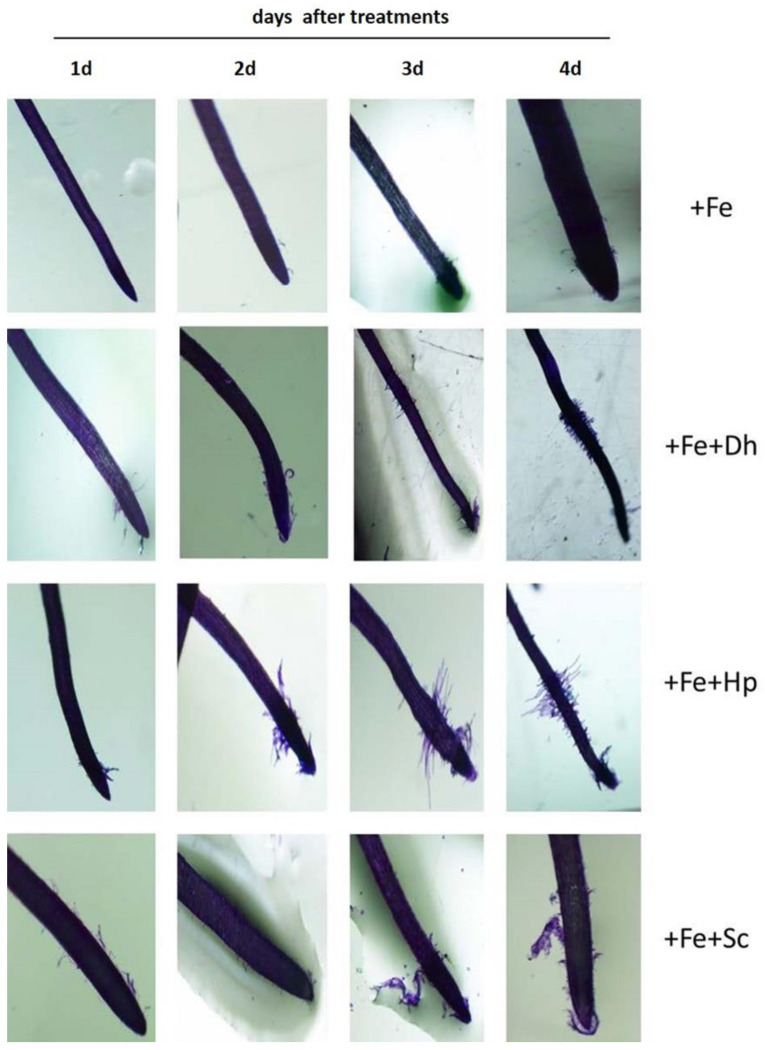
Analysis of development of subapical root hairs in young roots of cucumber plants grown with Fe (+Fe) or with Fe and inoculated with any of the three yeasts studied in this work: *Debaryomyces hansenii* (+Fe+Dh), *Hansenula polymorpha* (+Fe+Hp), or *Saccharomyces cerevisiae* (+Fe+Sc). The inoculation with the yeasts was carried out the same day the Fe-sufficient treatment was applied. Roots were observed 1, 2, 3, and 4 days after treatments (1d, 2d, 3d, and 4d). The images were obtained using a stereoscopic microscope with camera system after having stained the roots with toluidine blue for 10 min.

## Data Availability

The original contributions presented in the study are included in the article further inquiries can be directed to the corresponding author.

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
