# Peer review of "Several Yeast Species Induce Iron Deficiency Responses in Cucumber Plants (Cucumis sativus L.)"

_microorganisms, 2021, doi:10.3390/microorganisms9122603_

Round 1
Reviewer 1 Report
The MS Lucena et al. deals with the ability of some yeast to induce Fe deficiency root responses in cucumber.
The study is interesting and very well done.
I only see two points that might be interesting to explore.
(i) To investigate if these yeasts are able to induce Fe deficiency responses in roots of Fe sufficient plants.
(ii) To investigate the potential role of those hormones related to the control of these types of responses such as ethylene and IAA for instance.
Author Response
Dear reviewer,
The authors are really grateful for your kindly and good feedback of our paper.
“(i) To investigate if these yeasts are able to induce Fe deficiency responses in roots of Fe sufficient plants”.
In relation with this comment, we would like to say you that we tested the three yeasts in cucumber and tomato plants under Fe sufficient growth conditions. The most relevant results that we got was related to the root hair proliferation (already shown in Fig. 5) and commented in text. The effect over other response mechanisms to Fe deficiency was faint and no significant.
“(ii) To investigate the potential role of those hormones related to the control of these types of responses such as ethylene and IAA for instance”.
That is a really good idea. Long is known about the role of hormones related to the control of these types of responses (Romera et al., 2019; Lucena et al., 2019; García et al., 2021). This paper is a first report but in the next future we plan to get further details about that. Next publications of our group will be related with hormones and its role on the regulation of both mineral nutrition and benefit microorganisms.
Reviewer 2 Report
The research described in presented paper is related to agrobiotechnology in its broadest sense and is in line with the trends concerning the reduction of the use of chemicals in agriculture.
The subject of this publication is the study of plants response to conditions provoked by microorganisms colonizing the rhizosphere. Authors prospect this research as a way of solving iron deficiency in the soil and it is well known how important iron is for the metabolism of every cell. Such an assumption is not new because, it is well known from literature data, that root-associated microbes of various origin, can induce morphological and/or physiological responses to Fe deficiency in dicot plant species. Authors based their study on 3 particular species of yeasts, but did not explain what guided their choice of just these 3 model microorganisms - particularly surprising were the studies on S.cerevisiae and not surprisingly, these microorganisms generally failed to meet expectations. Definitely the manuscript should be enriched with the explanation of this issue. Besides, my doubts are raised by the publication of the manuscript in its current form in the journal as Microorganisms - for me it does not comply with its scope. To be considered for publication in mentioned Journal, it is necessary to change the proportions of the research and, in addition to data characterizing the plant response, it is necessary to provide information on the studied microorganisms and to try to investigate exactly how they act, what happens in their metabolism, etc. Or look for deeper dependencies in the results obtained on the species of yeast that gave the expected positive effect.
Author Response
Reviewer 2
Dear reviewer,
Thank you very much for your kindly comments. There is no doubt that they will help to improve our research. You can find our responses next to each of one, down.
“Such an assumption is not new because, it is well known from literature data, that root-associated microbes of various origin, can induce morphological and/or physiological responses to Fe deficiency in dicot plant species”.
Several studies, carried out with beneficial fungi or bacteria, have shown that its application to soils can improve Fe nutrition in plants (Romera et al., 2019). However, nothing is known up to date about the possible role that yeasts may play in the Fe nutrition of plants. Recently, Kaur et al., (2020) have characterized, for the first time, the role of certain strains of the yeast Debaryomyces hansenii in the detoxification of arsenic in rice plants. Furthermore, in that study the authors also observed that, apart from mitigating the stress caused by arsenic, inoculation with the yeast improved both the nutritional status and the growth of rice plants.
We think that it is, therefore, important to expand and optimize the list of possible beneficial microorganisms that can help plants to assimilate Fe and, in this way, to minimize the use of chemical fertilizers. Here, we describe and evaluate yeast species that have the capacity to induce Fe deficiency morphological and physiological responses and can help plants to acquire Fe.
“Authors based their study on 3 particular species of yeasts, but did not explain what guided their choice of just these 3 model microorganisms - particularly surprising were the studies on S.cerevisiae and not surprisingly, these microorganisms generally failed to meet expectations”.
We have added new information in the Introduction Section (lines 109-116). We explain the reasons to use the three yeasts used in this study. The yeast species analysed have been Debaryomyces hansenii, the nitrate Hansenula polymorpha and Saccharomyces cerevisiae. D. hansenii was selected because is an abundant yeast in soil and salty environments. In addition, it is a yeast very tolerant to several abiotic stress factors. H. polymorpha was chosen because, in addition of being abundant in soil, is a model to study mineral nutrition in plants since for example is a nitrate assimilating yeast. Finally, S. cerevisiae was used just because is a research model and most of the information in yeast has been used in Saccharomyces.
“Definitely the manuscript should be enriched with the explanation of this issue”.
We completely agree with you. As already mentioned, we have added the explanation of three yeasts selection into the introduction of the new version of our manuscript.
“Besides, my doubts are raised by the publication of the manuscript in its current form in the journal as Microorganisms - for me it does not comply with its scope”.
We can understand your concern about the scope. The main reason why we decided submitting the manuscript to this journal was the special issue´s title: “Exploring Fungal Diversity: Novel Bioactive Compounds and Sustainable Bioprocesses”. The authors humbly considered that our researching is very related with the role that microorganisms play on the Sustainable Bioprocesses, considering the use of these microorganisms as agrobiotechnological tool concerning the reduction of the use of chemicals in agriculture.
In fact, the editor did not mind accepting the abstract of our research for submitting into of its special issue
“To be considered for publication in mentioned Journal, it is necessary to change the proportions of the research and, in addition to data characterizing the plant response, it is necessary to provide information on the studied microorganisms and to try to investigate exactly how they act, what happens in their metabolism, etc”.
Once again, the reviewer is completely right. In this research we show that some yeast may play an important role inducing morphological and/or physiological responses to Fe deficiency in dicot plant species. Nothing is known up to date about that. Ours, is the first research that demonstrate that yeasts can induce these kind of response mechanisms. In the next future we plan to get further details about the physiological and molecular reasons behind the positive effect observed in this paper.
Round 2
Reviewer 2 Report
Authors have made a partial correction of the text according to my indications. But nothing has changed in the context of the most important objection related to the improper selection of the type of journal for the material described. Even taking into account the special issue they write about. The microbiological section is still very poor and is limited to just pointing out that the tested yeast may or may not affect the plant. The authors explain that they are going to start research on yeasts in the future. In this case, the solution is simple: either publishing the manuscript in its present form in a more suitable journal, or enriching it with at least basic research on microorganisms and publishing it in Microbiology.
Author Response
We are ataching a file with our answer.
We consider the manuscript fits with the scope of the special issue
